# A Review of Medicinal Plants with Renoprotective Activity in Diabetic Nephropathy Animal Models

**DOI:** 10.3390/life13020560

**Published:** 2023-02-16

**Authors:** I Made Wisnu Adhi Putra, Nanang Fakhrudin, Arief Nurrochmad, Subagus Wahyuono

**Affiliations:** 1Department of Biology, University of Dhyana Pura, Badung 80351, Indonesia; 2Doctorate Program of Pharmacy, Universitas Gadjah Mada, Yogyakarta 55281, Indonesia; 3Department of Pharmaceutical Biology, Faculty of Pharmacy, Universitas Gadjah Mada, Yogyakarta 55281, Indonesia; 4Medicinal Plants and Natural Products Research Center, Faculty of Pharmacy, Universitas Gadjah Mada, Yogyakarta 55281, Indonesia; 5Department of Pharmacology and Clinical Pharmacy, Faculty of Pharmacy, Universitas Gadjah Mada, Yogyakarta 55281, Indonesia

**Keywords:** antidiabetic activity, medicinal plants, renal, phytotherapy

## Abstract

Diabetic nephropathy (DN), also recognized as diabetic kidney disease, is a kidney malfunction caused by diabetes mellitus. A possible contributing factor to the onset of DN is hyperglycemia. Poorly regulated hyperglycemia can damage blood vessel clusters in the kidneys, leading to kidney damage. Its treatment is difficult and expensive because its causes are extremely complex and poorly understood. Extracts from medicinal plants can be an alternative treatment for DN. The bioactive content in medicinal plants inhibits the progression of DN. This work explores the renoprotective activity and possible mechanisms of various medicinal plant extracts administered to diabetic animal models. Research articles published from 2011 to 2022 were gathered from several databases including PubMed, Scopus, ProQuest, and ScienceDirect to ensure up-to-date findings. Results showed that medicinal plant extracts ameliorated the progression of DN via the reduction in oxidative stress and suppression of inflammation, advanced glycation end-product formation, cell apoptosis, and tissue injury-related protein expression.

## 1. Introduction

Diabetes mellitus (DM), a metabolic disorder, is defined as the body’s inability to produce or effectively retaliate to insulin [1]. DM affected 463 million people worldwide in 2021. The amount of people suffering from diabetes is estimated to increase to 578 million in 2030 and 700 million in 2045 [2]. DM is identified by elevated blood glucose levels and a number of complications [3]. Diabetic nephropathy (DN) is a DM complication that affects roughly one-third of all patients with diabetes [4]. The clinical features of DN include increased albumin excretion in the urine (300 mg/day), low rate of glomerular filtration, susceptibility to persistent hyperglycemia throughout the pre-diabetic period, and gradual reduction in kidney function [5]. DN has become the most frequent source of end-stage renal disease (ESRD), which eventually leads to kidney damage [6].

The causative factors of DN are extremely complex and poorly understood; thus, the treatment strategy is difficult and expensive. Current advances in DN medication have failed to improve patient outcomes [7]. Moreover, the use of multiple synthetic medications has decreased patient compliance and resulted in a variety of adverse effects, ultimately worsening patients’ condition [8,9]. Therefore, alternative strategies for preventing or treating DN are urgently needed. Herbal medicines have bioactive constituents that can treat diabetes and its complications and are safer and more effective than synthetic medicines [10,11].

Reviews on the renoprotective effects of medicinal plants in diabetic animal models were very limited. Therefore, the aim of this work was to explore the research findings related to medicinal plants and their mechanisms of action in treating or preventing DN. In addition, the phytochemical content in medicinal plants that are responsible for renoprotective activity were also described.

## 2. Data Sources and Search Strategy

All of the articles in this review that discuss medicinal plants used in the treatment of DN were found online. We collected articles published from 2011 to 2022 from several databases, namely, PubMed, Scopus, ProQuest, and ScienceDirect, using the following keywords: “Renoprotective activity”, “Extract”, “Diabetic nephropathy”, and “Diabetic kidney disease”. Medicinal plant extracts showing an effect on DN in preclinical studies (in vivo) were the focus of our search. Furthermore, we only included articles discussing the renoprotective effect of a single extract. Among the obtained articles, we excluded those presenting combined extracts, isolated compounds, unclear doses, and fermented products and those not explaining the mechanism.

## 3. Diabetic Nephropathy

DM causes a variety of serious complications, of which DN is a serious health problem worldwide and occurs in about 40% of people with diabetes [10]. DN increases the mortality rate of patients with diabetes, although deaths from complications of diabetes are mostly caused by cardiovascular problems [12]. DN has five stages, namely, hyperfiltration, normoalbuminuria, microalbuminuria, macroalbuminuria, and ESRD [13]. ESRD may occur when DN ultimately results in kidney damage [9] via hyperfiltration, increased albumin excretion, microalbuminuria, nodular and diffuse glomerulosclerosis, and proteinuria. Furthermore, chronic hyperglycemia causes structural abnormalities including tubular atrophy, glomerular basement membrane-stiffening, kidney enlargement, hypertension, edema, and progressive renal dysfunction in the later stages of ESRD. The number of patients with ESRD increases with DN cases. Therefore, dialysis or kidney transplantation is urgently needed for patient survival [14].

DN is characterized by the structural abnormalities of kidney tissues, such as glomerular enlargement, advancement of mesangial matrix, and stiffening of tubular and glomerular basement membranes. Patients with DN have abnormal levels of urea, albumin, creatinine, uric acid, and blood urea nitrogen (BUN) in their serum and experience fluid retention, glomerular lesions, and glomerular filtration rate (GFR) reduction [15,16]. Several causative factors of DN have been proposed, including oxidative stress enhancement, mitogen-activated protein kinases, polyol pathways, protein kinase C (PKC), and poly(ADPribose) polymerase activation, advanced glycation end product (AGE) production, inflammatory mediators (TNF-α, IL-2, IL1β, and IL-6), growth factors (VEGF and CTGF), and chemokine liberation. Among these factors, oxidative stress is believed to be a factor in the onset and progression of DN [17].

DN pathogenesis is incredibly complex and remains largely unknown; it implicates the direct action of excessive extracellular glucose in tubular, glomerular, interstitial, and vascular cells. As a result, the therapeutic outcomes for DN are poor [18]. Standard medication, including stringent blood pressure and sugar control, seems to be ineffective in arresting the development of DN to ESRD and DN-related mortality. Another treatment strategy for DN is the administration of insulin and antidiabetic drugs, including sulfonylureas, biguanides, thiazolidinediones, insulin sensitizers, inhibitors of α-glycosidase, agonists of glucagon-like peptide, inhibitors of dipeptidyl peptidase-4, incretin-based drugs, and inhibitors of sodium-glucose cotransporter. These drugs are usually administered with attention to the GFR of the patient [19]. However, this strategy only slows the disease’s progression. Thus, the discovery of novel medications targeting DN pathology including oxidative stress and inflammation has taken center stage [20].

## 4. Animal Model of DN

Several animals are often used as models for DN, and rodents are one of the most common. Rats and mice are two species of rodents commonly employed as animal models in preclinical studies for metabolic disorders, including DN, because of their human-like physiology, handling simplicity, and housing convenience [21]. The use of these animals can provide insight into the mechanisms and effects of medicinal plant extracts on individuals with DN. Diabetogenic agents, including streptozotocin, alloxan, a high fat diet (HFD), a fructose diet, and combinations of these ingredients, are used to induce DM in animals and establish DN conditions measured by several biomarkers, including the increase in serum urea, serum creatinine, serum uric acid, BUN, total urinary protein, and urinary albumin [22,23].

### 4.1. DN Animal Model Induced by Streptozotocin

Streptozotocin (STZ), an analog of N-acetylglucosamine (GlcNAc), easily enters pancreatic β-cells via the glucose transporter-2 (GLUT-2), resulting in β-cell toxicity and insulin deficiency [24,25]. O-GlcNAcase, which eliminates O-GlcNAc from proteins, is specifically inhibited by STZ, resulting in β-cell apoptosis and persistent intracellular protein O-glycosylation [26]. STZ can cause hyperglycemia by selectively damaging pancreatic β-cells. An increase in STZ dosage results in increased cytotoxicity and greater pancreatic β-cell destruction, which induces and intensifies DM [27]. A large dose of STZ is harmful to pancreatic β-cells and causes severe insulin deficiency—a characteristic of type 1 DM [28]. In addition, a high STZ dose has a toxic effect on the kidneys of animal models by producing acute injury. Mice treated with large dosages of STZ may develop nephropathy, an injury caused by hyperglycemia, on top of acute renal STZ cytotoxicity [23,29]. Within 48 h of receiving a single injection of a high STZ dose, full β-cell damage and DM could occur. Meanwhile, STZ administered at low doses for 5 days causes partial β-cell loss, resulting in hypoinsulinemia and hyperglycemia [30]. In rats, the early phases of STZ-induced DM features oxidative stress elevation and mitochondrial dysfunction. This finding is evidence that the development of DM is initiated by oxidative stress. Given that STZ damages pancreatic β-cells in rats and nicotinamide only partially protects pancreatic β-cells from STZ, administering STZ in combination with nicotinamide in rats could promote type 2 DM [31,32].

### 4.2. DN Animal Model Induced by Alloxan

Alloxan is a common diabetogenic substance used to induce DM in animal models. It is an organic molecule from urea and has a cytotoxic effect on pancreatic β-cells. Alloxan raises the level of reactive oxygen species (ROS) and facilitates the selective necrosis of pancreatic β-cells [33]. When administered parenterally (intravenously, intraperitoneally, or subcutaneously), this substance exhibits a diabetic effect. Persistent DM is promoted by alloxan via pancreatic islet β-cell damage in a variety of species [34]. This compound has minimal effect on other tissues when administered at the proper dose and is selective to islet β-cells. DM induction by alloxan involves the partial deterioration of pancreatic islet cells, which results in a decrease in the quantity and quality of insulin. Alloxan selectively inhibits the secretion of insulin by blocking the activity of glucokinase, the glucose sensor of β-cells. This compound also suppresses glucokinase by reacting with two -SH groups in the enzyme’s sugar-binding region [33,35]. Alloxan generates insulin-dependent diabetes by triggering the formation of ROS, resulting in β-cell necrosis. It penetrates β-cells, participates in a number of processes that could harm β-cells and induce necrosis, and generates ROS in a cyclic reaction with dialuric acid, its reduction product. The toxicity of alloxan is facilitated by the ROS created during this redox reaction [36].

### 4.3. DN Animal Model Induced by High Fat Diets (HFDs)

HFDs, which are foods with a high fat content, are considered a diabetogenic agent. In diabetic models induced by HFD, ROS generation may be the initial event that leads to insulin resistance [37,38]. HFD in combination with low-dose STZ has been used to immediately induce type 2 DM in rats [39]. HFD administration is valuable for examining the mechanisms behind insulin resistance; however, animals rarely exhibit overt hyperglycemia and do not display the traditional symptoms of human DN, and a HFD may itself induce renal injury [40].

### 4.4. Specific Genetic Rodent Models

Specific genetic rodent models including leptin-null (ob/ob) mice and leptin receptor (db/db) mutant mice have been widely used as preclinical models for DM. In recent years, the pathophysiology of DN has been studied using C57BL/KsJ-db/db mice as a spontaneous diabetic model of type 2 DM. C57BL/KsJ-db/db mouse is a well-known model of obesity-induced type 2 DM and carries a mutation in the leptin receptor gene [41].

## 5. Bioactive Phytochemicals for DN

Medicinal plants have many chemical compounds or phytochemicals that have various pharmacological activities [42]. The use of phytochemicals and their bioactive components as natural modifying agents in the cure of numerous ailments, including DN, is supported by growing evidence of their positive health effects [43]. Medicinal plants contain major phytochemical compounds with health advantages [44], including alkaloids, phenolics, flavonoids, terpenoids, and miscellaneous compounds. These compounds are important in ameliorating DN progression.

### 5.1. Alkaloids

Alkaloids are basic chemical compounds (containing nitrogen) present in a wide range of species, such as fungi, bacteria, plants, and animals. They are found in ca. 300 plant families [45]. The essential feature of alkaloids is their heterocyclic ring structure that contains a nitrogen atom (others are aliphatic nitrogen compounds—noncyclic) [46]. Alkaloids can exist as homo- or hetero-oligomeric monomers, dimers, trimers, or tetramers. They are divided into two groups: those with heterocyclic and nonheterocyclic chemical structures and those with biological or natural origins (specific sources). To date, more than 20,000 alkaloids have been isolated, of which about 600 have been studied for their biological effects [47]. Figure 1 shows some alkaloid compounds that have exhibited renoprotective activity in DN animal models. All of these alkaloids have been studied preclinically for their activity in ameliorating DN progression. They are berberine, magnoflorine, trigonelline, and oxymatrine.

#### 5.1.1. Berberine

Berberine is an isoquinoline alkaloid quaternary ammonium salt found in plants and comes from a wide range of plants, including *Arcangelisia flava*, *Berberis aquifolium*, *Berberis aristata*, *Berberis vulgaris,* and *Hydrastis canadensis* [48]. The protective effect of berberine on the kidney was observed by some researchers. This compound protects the kidney from injury by blocking TGF-β/Smad3-mediated renal fibrosis and NF-κB-induced renal inflammation [49]. It lessens renal injury, inflammation, and podocyte death by hindering the TLR4/NF-B pathway [50]. Furthermore, berberine reduces renal tubulointerstitial fibrosis by preventing hyperglycemia-induced EMT through NLRP3 inflammasome inactivation [51].

#### 5.1.2. Magnoflorine

Magnoflorine is a significant quaternary aporphine alkaloid found in the roots, rhizomes, stems, and bark of several medicinally significant plants, including *Berberis kansuensis* C.K.Schneid., *Magnolia officinalis* Rehder & E.H. Wilson, *P. amurense*, *S. acutum*, and *Thalictrum isopyroides* C.A. Mey. [52]. This metabolite attenuates DN and suppresses inflammatory responses and fibrosis by facilitating the consistent production of lysine-specific demethylase 3A (KDM3A) [53].

#### 5.1.3. Oxymatrine

Oxymatrine is a quinolizidine alkaloid found naturally in *Sophora flavescens* Ait roots and has strong pharmacological effects [54]. In diabetic rats, oxymatrine considerably lowers oxidative stress and the levels of *transforming growth factor-*β1 (TGF-β1), AGEs, CTGF, and inflammatory cytokines in the kidney [55]. By increasing Id2 expression, oxymatrine also prevents twist-mediated renal tubulointerstitial fibrosis [56].

#### 5.1.4. Trigonelline

Trigonelline is a polar/hydrophilic pyridine alkaloid derived from a variety of plant species, including *Trigonella foenum graecum*, *Allium sepapea*, *Coffea sp*, *Pisum sativum*, *Glycine max,* and *Lycopersicon esculentum* [57]. This compound suppresses renal oxidative stress and reduces renal cell apoptosis and fibrosis by regulating the PPAR-γ/GLUT4-leptin/TNF-α signaling pathway [58].

### 5.2. Phenolics

Phenolic compounds comprise a group of compounds (secondary metabolites) generated by the phenylpropanoid metabolization of plant shikimic acid and pentose phosphate. Their structure consists of benzene rings with hydroxyl substituents, and their complexity ranges from simple to complex polymers of phenolic compounds [59]. Phenolics, which are the most noticeable secondary metabolites in plants, are distributed throughout the entire metabolic process and comprise a wide range of compounds, including phenolic acids, flavonoids, anthocyanins, stilbenes, and lignans [60]. In general, these molecules play an important role in plant defense mechanisms, such as against infection or insect infestation, ultraviolet light, and mechanical injury. They are also important in human health because they protect against ROS damage [61]. Figure 2 shows several phenolic compounds that have been reported to have renoprotective activity in experimental animal models of DN. They are mangiferin, syringaresinol, resveratrol, and gallic acid. These compounds regulate signaling cascades including those engaged in response to oxidative stress, anti-inflammatory, and apoptosis [62].

#### 5.2.1. Resveratrol

Resveratrol is a phytoalexin that occurs naturally by specific spermatophytes when they are injured [63]. Numerous animal studies showed that resveratrol can help manage diabetes through a variety of mechanisms. Resveratrol ameliorates DN progression by inhibiting nuclear factor-κB (NF-κB) activity and attenuating renal mesangial cell proliferation [64], lowering oxidative stress and downregulating receptor for advanced glycation end product (RAGE) expression [65], lowering AGE accumulation, oxidative damage, apoptosis, and NADPH oxidase 4 (NOX4) expression [66,67].

#### 5.2.2. Gallic Acid

Gallic acid is phenolic compound belonging to phenolic acid subclasses. It downregulates the renal expression of TGF-β1 in diabetic Sprague–Dawley rats [68] and reduces microRNA-associated fibrosis and endoplasmic reticulum (ER) stress and elevates glyoxalase 1 (GLO1) activity and Nrf2 adjustment in diabetic NMRI mice [69].

#### 5.2.3. Syringaresinol

Syringaresinol is a lignan that occurs naturally in plants such as flax seed, sesame seed, Brassica vegetables, and grains [70]. Its ameliorating activity for the progression of DN is observed in diabetic rats and occurs through the activation of Nrf2 and the inactivation of TGF-β1/Smad pathways [71]. In diabetic C57BL/6 mice, syringaresinol inhibits pyroptosis by activating the Nrf2 antioxidant pathway [72].

#### 5.2.4. Mangiferin

Mangiferin is a xanthone that can be found at large concentrations in higher plants and in the peel, stalks, leaves, barks, kernels, and stones of the mango fruit. It has a broad range of therapeutical effects, including antioxidant and antidiabetic properties [73]. In the treatment of DN, mangiferin reduces oxidative stress and cell apoptosis in diabetic rats [74]. Mangiferin also protects podocytes in diabetic rats by increasing autophagy via the AMPK-mTOR-ULK1 pathway [75].

### 5.3. Flavonoids

Among plant secondary metabolites, flavonoids are the most diverse groups. They are a type of phenolics with a flavone backbone (2-phenylchromen-4-one) and are present in many different foods and beverage items. The functional groups attached to the basic flavonoid structure define the subclasses of flavones, isoflavones, flavanols, flavonols, flavanones, and anthocyanins. Flavonoids have a diverse range of pharmacological activities. Various investigations have been conducted on their significant role in diabetes’ treatment and its complications [76,77]. Several flavonoids including: hesperetin, luteolin, catechin, genestein, quercetin, and kaempferol are involved in the treatment of DN, and their structures are depicted in Figure 3.

#### 5.3.1. Hesperetin

Hesperetin belongs to the flavanones’ class of flavonoids and is one of the most abundant flavonoids discovered in citrus fruits [78]. This compound has a wide range of pharmacological effects, including the ability to slow the progression of DN. Chen et al. [79] studied the effect of hesperetin on diabetic Sprague–Dawley rats and found that this compound ameliorated the progression of DN via upregulating Glo-1, inhibiting the AGE/RAGE axis and inflammation, elevating Nrf2 and p-Nrf2 levels, and upregulating γ-glutamylcysteine synthetase.

#### 5.3.2. Luteolin

Luteolin is a flavone that occurs naturally as a glycosylated form in a variety of fruits and vegetables. In ameliorating the DN progression of diabetic Sprague–Dawley rats, luteolin protects the basement membrane’s filtration function by increasing Nphs2 protein expression and halting the apoptosis, removal, and integration of podocytes [80]. In C57BL/6 J db/db and C57BL/6 J db/m mice, luteolin suppresses inflammatory response and oxidative stress by inhibiting the signal transducer and activator of transcription 3 (STAT3) pathway [81].

#### 5.3.3. Catechin

Catechin is a flavanol and a secondary metabolite with antioxidant roles in plants. Investigation on db/db mice revealed that catechin inhibits AGE formation and disconnects the inflammatory pathway via methylglyoxal trapping [82]. In diabetic rats, catechin elevates the expression of liver glucose-metabolism enzymes and proteins related to insulin signal-transduction pathways [83].

#### 5.3.4. Genistein

Genistein is an isoflavone flavonoid and primarily a phytoestrogen obtained from legumes [84]. A study on diabetic mice showed that genistein inhibits renal fibrosis by blocking the formation of fibrosis-related markers [85]. In diabetic Sprague–Dawley rats, genistein improves renal fibrosis by regulating the TGF-1/Smad3 pathway and reduces oxidative stress by activating the Nrf2-HO-1/NQO1 pathway [86].

#### 5.3.5. Quercetin

Quercetin is a compound belonging to the flavonol subclasses of flavonoid. In diabetic rats, quercetin suppresses the oxidative stress by regulating TGF-β1 expression [87]. Investigation on C57BL/KSJ db/db mice showed that quercetin attenuated podocyte apoptosis by inhibiting the EGFR signaling pathway [88]. In diabetic mice, quercetin ameliorates renal fibrosis by upregulating the autophagy level mediated by beclin-1 and downregulating snail1 expression [89].

#### 5.3.6. Kaempferol

Kaempferol is a flavonol having a chemical structure similar to quercetin [90]. The activity of kaempferol in treating DN has been investigated. In diabetic C57BL/6 mice, kaempferol ameliorates renal fibrosis by inhibiting fibrogenesis through Rhoa/Rho kinase [91]. Another investigation revealed that kaempferol reduces the oxidative stress of diabetic rats by activating the Nrf-2/HO-1/antioxidant axis [92].

### 5.4. Terpenoids

Terpenoids, sometimes referred to as isoprenoids, are a group of chemical compounds derived from isoprene that are widely distributed throughout various plant, fungal, algal, and sponge taxa. Owing to their wide range of medical applications, these compounds have achieved an important pharmacological value [93]. Despite being defined as an altered group of terpenes with different functional groups and oxidized methyl groups excluded at different locations, terpenes and terpenoids are not the same but sometimes used interchangeably [94]. Terpenoids are easily categorized according to the number of their carbon atoms as follows: hemiterpenoids, monoterpenoids, homoterpenoids, sesquiterpenoids, diterpenoids, sesterpenoids, triterpenoids, tetraterpenoids, and polyterpenoids with carbon atoms of C_5_, C_10_, C_11,16_, C_15_, C_20_, C_25_, C_30_, C_40_, and C_>40_, respectively [95]. Terpenoids may produce protective effects on DN animal models. Some examples of these compounds are listed in Figure 4. They are andrographolide, astragaloside IV, lupeol, and ursolic acid.

#### 5.4.1. Andrographolide

Andrographolide is a labdane diterpenoid generated by *Andrographis paniculata* with a wide range of medicinal applications, including antidiabetic [96]. In the treatment of DN, this compound reduces hyperglycemia-induced renal oxidative stress and inflammation via the Akt/NF-κB pathway [7].

#### 5.4.2. Astragaloside

Astragaloside IV, the active component of *Astragalus membranaceus,* is a tetracyclic triterpenoid saponin derived from lanolin alcohol [97] and has shown strong protective effects on the kidneys of DN animal models. The renoprotective effect of astragaloside IV may be mediated by the repair of the mitochondrial quality control network, which can reduce renal damage in db/db mice [98], and the inhibition of NLRP3 inflammasome-mediated inflammation [99].

#### 5.4.3. Lupeol

Lupeol is a common natural triterpenoid that can be found in a variety of plants (including licorice and *Emblica officinalis*), fruits (including mango and strawberry), and vegetables (including white cabbage and pepper) [100]. It exhibits renoprotective activity in the DN animal model by increasing the activity of antioxidant enzymes (GSH, CAT, and SOD), which in turn reduce oxidative stress [101].

#### 5.4.4. Ursolic Acid

Ursolic acid is a naturally occurring pentacyclic triterpenoid found in a variety of medicinal plants including hawthorn, bearfruit, and Fructus Ligustri. Owing to its various pharmacological activities, this compound has sparked considerable interest in recent years [102]. In DN animal models, ursolic acid suppresses oxidative stress and inflammation [103], inhibits extracellular matrix accumulation and renal fibrosis [104], and inhibits AGEs’ formation in kidney and plasma [105].

### 5.5. Miscellaneous Compounds

In addition to the abovementioned secondary metabolites, some other classes of plant-derived compounds have beneficial effects on DN, including allicin and 4-hydroxyisoleucine. The chemical structure of these compounds is depicted in Figure 5.

#### 5.5.1. Allicin

Allicin or diallyl thiosulfinate is the most abundant thiol-reactive organosulfur compound generated by *Allium sativum* in response to tissue damage [106]. Allicin shows its renoprotective effect in DN animal models by ameliorating the diabetes-induced morphological alterations of the kidney [107] and suppressing oxidative stress and renal inflammation [108]. It also slows down nephropathy progression by inhibiting the OS-hypoxia-fibrosis pathway, HIF-1, and CTGF [109].

#### 5.5.2. 4-Hidroxyisoleucine

4-Hydroxyisoleucine is an amino acid found only in plants, particularly in fenugreek, and is not found in mammalian tissues. Fenugreek, a leguminous plant, has been used in traditional medicine to treat diabetes, and one of its active components is 4-hydroxyisoleucine [110]. In the DN animal model, 4-hydroxyisoleucine decreases the inflammatory response and oxidative stress through the TGF-β1 signaling pathway and Nrf2 gene activation [111].

## 6. Renoprotective Mechanisms of Medicinal Plants

Phytochemicals derived from medicinal plants are less harmful than medications made from synthetic substances. However, their precise bioactive components, mode of action, pharmacological properties, and potential risks are poorly understood [8]. Plant extracts exhibit renoprotective activity in various ways. As shown in Figure 6, the renoprotective activity of medicinal plant extracts are realized by reducing oxidative stress and suppressing inflammation, AGE production, cell apoptosis, and tissue injury-related protein expression. The complete information of the plants’ scientific names, extract forms, animal models, extract doses, and possible modes of action are presented in Table 1.

### 6.1. Reduction in Oxidative Stress

Oxidative stress is one of the main determinants involved in the onset and progression of DN pathogenesis [112]. This condition is promoted by hyperglycemia through the increased production of ROS, which alters the metabolism of carbohydrates and causes complications including DN [113,114]. Oxidative stress-related renal damage in DN is characterized by significant structural and functional alterations in glomerular and renal tubular cells. The three main sources of ROS in DM are NOX, AGE, and polyol chain. In particular, the NOX4 (family of NOX) enzyme is critically necessary for the kidneys to produce ROS. Furthermore, ROS is produced via NOX, uncoupled nitric xanthine oxidase, oxide synthase, lipoxygenase, and mitochondrial respiratory chain dysfunction [20,115]. In general, renal cells have a self-defense system that protects them from ROS. This system is known as an antioxidant, which can be enzymatic (superoxide dismutase (SOD), catalase (CAT), and glutathione peroxidase (GPx)) or non-enzymatic (GSH). In hyperglycemic conditions, the amount of produced ROS may exceed the amount that renal antioxidants can control [116,117]. The imbalance in the ROS: antioxidant ratio causes changes in the redox signaling of the cell, which in turn leads to an impairment in the cell metabolism [118]. In addition, multiple pathways involved in DN pathogenesis are induced by this imbalance. As a result, antioxidative-stress treatment strategies may efficaciously maintain normal renal function while halting or delaying DN progression [119].

Natural antioxidants can attenuate oxidative stress by inhibiting the formation of ROS, scavenging and inactivating ROS, inducing the activity of antioxidant enzymes, and forming other proteins involved in the antioxidant pathway [120]. Plant extracts exhibit antioxidant activity in experimental animal models of DN. From Table 1, we can see that the water extract of the seeds of *Trigonella foenum graecum* counteracts the free radicals and reduces the renal damage in high-sucrose diet and STZ (25 mg/kg) induced diabetic Sprague–Dawley rats [121]. In diabetic Wistar rats induced by STZ (30 mg/kg), the water extract of the fruit pulp of *Passiflora ligularis* Juss decreases the oxidative stress [122]. In this research, the therapy was conducted for 30 days at doses of 200 mg/kg, 400 mg/kg, and 600 mg/kg orally per day. The same result was also obtained by Atawodi et al. [123], who administered the methanol extract of the leaves of *Tetrapleura tetraptera* to diabetic Wistar rats induced by alloxan (120 mg/kg) and boosted 4 days later (120 mg/kg). Furthermore, the methanol extract of the pods of *Acaciella angustissima* (25 mg/kg, 50 mg/kg, and 100 mg/kg) showed antioxidant activity by reducing renal TBARS in diabetic Wistar rats induced by STZ (45 mg/kg) [124]. The extract was given at a dose of 50 mg/kg orally per day for 7 days.

Medicinal plant extracts exhibit antioxidant activity by restoring the enzymatic antioxidative defense system. Dogan et al. [125] administered the water extract of the leaves of *Quercus brantii* with doses 100 mg/kg, 250 mg/kg, and 500 mg/kg to diabetic Wistar rats induced by 50 mg/kg of STZ (Table 1). After 21 days, a decrease in renal GSH, GST, CAT, GPx, and SOD was observed in the treatment group compared with those in the diabetic control. The water extract of the leaves of *Cyclocarya paliurus* (47 and 94 mg/kg orally per day for 56 days) improved the renal CAT, GPx, and SOD in diabetic Wistar rats induced by HFD and 35 mg/kg of STZ [126]. The ethanolic extract of the stem barks of *Ficus recemosa* (200 mg/kg and 400 mg/kg orally per day for 56 days) restored the renal GSH and SOD in diabetic Wistar rats induced by 45 mg/kg of STZ [127]. The ethanolic extract of the flowers of *Diplotaxis simplex* (100 mg/kg and 200 mg/kg orally per day for 30 days) improved renal SOD, CAT, and GPx in diabetic Wistar rats induced by 150 mg/kg of alloxan [128]. The methanolic extract of the barks of *Syzygium mundagam* (100 mg/kg and 200 mg/kg orally per day for 28 days) increased the renal SOD, CAT, GSH, and GST in diabetic Wistar rats induced by 60 mg/kg of STZ and 120 mg/kg of nicotinamide [129].

Other findings were also reported in the restoration of the enzymatic antioxidative defense system (Table 1). These treatments include the methanolic extract of the leaves of *Anogeissus acuminata* (100 mg/kg and 300 mg/kg for 56 days) administered to diabetic Wistar rats induced by 50 mg/kg of STZ [130]; the ethanolic extract of the buds and flowers of *Cassia auriculata* (250 mg/kg and 500 mg/kg orally per day for 21 days) administered to diabetic Wistar rats induced by HFD and 35 mg/kg of STZ [131]; the ethanolic extract of the stems and roots of *Nerium oleander* (200 mg/kg orally per day for 20 days) administered to diabetic Swiss albino mice induced by 150 mg/kg of alloxan [132]; the water extract of the leaves of *Nelumbo nucifera* (0.5% and 1% (*w*/*w*) orally per day for 42 days) administered to diabetic Sprague–Dawley rats induced by HFD and 35 mg/kg of STZ [119]; the hydroethanolic extract of the aerial parts of *Ficus religiosa* (50 mg/kg, 100 mg/kg, and 200 mg/kg orally per day for 45 days) administered to diabetic Wistar rats induced by 65 mg/kg of STZ and 230 mg/kg of nicotinamide [133]; the methanolic extract of the aerial parts of *Phyllanthus fraternus* (200 mg/kg and 400 mg/kg orally per day for 14 days) administered to diabetic Wistar rats induced by 130 mg/kg of alloxan [134]; the aqueous extract of the flower of *Etlingera elatior* (1000 mg/kg orally per day for 42 days) administered to diabetic Sprague–Dawley rats induced by HFD and 35 mg/kg of STZ [135]; the methanolic extract of the aerial part *Centaurium erythraea* (100 mg/kg orally per day for 28 days) administered to diabetic Wistar rats induced by STZ (40 mg/kg) for 5 consecutive days [136]; and the hydroalcoholic extracts of the aerial part of *Thuja occidentalis* (50 mg/kg, 100 mg/kg, and 200 mg/kg orally per day for 30 days) administered to diabetic Wistar rats induced by 65 mg/kg of STZ and 230 mg/kg of nicotinamide [137]. In general, these findings indicate the ability of plant extracts to reduce malondialdehyde (MDA) levels in the kidneys of DN models. MDA is a marker compound that shows cellular damage due to oxidative stress [138]. Animal models with DN have higher MDA levels than normal animal models [139].

**Table 1 life-13-00560-t001:** Medicinal plants with renoprotective activity in diabetic nephropathy animal models.

Plant Name	Extract Form	Animal, Induction	Extract Dose/Day	Duration	Observed Effects	References
*Cornus officinalis*	Water extract of the fruits	C57BL/KsJ-db/db mice	500 mg/10 mL/kg	56 days	↑ Renal SOD ↓ Renal XO, CAT, GST, eNOS	[140]
*Salvia miltiorrhiza*	Water extract of the roots	Sprague–Dawley rats, STZ	500 mg/kg	56 days	↓ 24-h urine protein ↓ BUN ↓ TGF-β1, collagen IV, ED-1, and RAGE	[141]
*Angelica Acutiloba*	Hydroalcoholic extract of the roots	Wistar rats, STZ	50 mg/kg, 100 mg/kg, 200 mg/kg	56 days	↓ Plasma glucose ↓ Serum and renal creatinine ↓ Urine volume ↓ Renal AGEs ↓ Mitochondrial TBARS	[142]
*Trigonella foenum graecum*	Water extract of the Seeds	Sprague–Dawley rats, sucrose enriched diets and STZ	400 mg/kg, 870 mg/kg, 1740 mg/kg	42 days	↓ Blood glucose ↓ BUN ↓ Serum creatinine ↓ urinary protein ↓ 8-OHdG ↓ Renal damage ↑ Serum and renal SOD, CAT ↓ Serum and renal MDA	[121]
*Smallanthus sonchifolius*	Water extract of the leaves	Wistar rats, STZ	70 mg/kg	28 days	↓ Blood glucose ↓ Urine volume ↓ Creatinine clearance ↓ Renal damage ↓ TGF-β1	[143]
*Cornus officinalis* SIEB. et Zucc.	Ethanolic extract of the fruits (*Corni Fructus*)	Wistar rats, STZ	100 mg/kg, 200 mg/kg or 400 mg/kg	40 days	↓ Blood glucose ↓ Urinary protein ↓ Serum albumin ↓ BUN ↓ Serum creatinine ↑ Renal CAT, SOD, GPx ↑ PPARγ expression ↓ Renal damage	[144]
*Portulaca oleracea*	Water extract of the aerial parts	C57BL/KsJ-db/db mice	300 mg/kg	70 days	↓ Blood glucose ↓ Plasma creatinine ↓ Urine volume ↓ Renal TGF-β1, AGEs ↓ ICAM-1 expression ↓ NF-κB p65 activation	[145]
*Liriope spicata* var. *prolifera*	Aqueous–ethanol extract of the tuberous roots	Wistar rats, STZ	100 mg/kg, 200 mg/kg	56 days	↓ Blood glucose ↓ Urine volume ↓ BUN ↓ Serum creatinine ↑ Creatinine clearance ↓ Renal damage ↓ TNF-α and IL-1β ↓ ICAM-1, MCP-1, fibronectin ↑ IκBα ↓ NF-κB	[146]
*Zingiber zerumbet*	Ethanolic extract	Wistar rats, STZ	200 mg/kg, 300 mg/kg	56 days	↓ Blood glucose ↓ BUN, serum creatinine ↓ Urine volume, proteinuria ↑ Renal nephrin and podocin ↑ pAMPK/AMPK ratio	[147]
*Passiflora ligularis* Juss.	Aqueous extract of the fruit pulp	Wistar rats, STZ	200 mg/kg, 400 mg/kg, 600 mg/kg	30 days	↓ Blood glucose ↑ Renal protein ↓ Serum creatinine ↓ Serum urea ↑ Renal SOD, CAT, GSH	[122]
*Tetrapleura tetraptera*	Methanol extract of the leaves	Wistar rats, alloxan	50 mg/kg	7 days	↓ Blood glucose ↓ Serum urea and creatinine ↓ MDA ↑ Renal SOD, CAT, GSH	[123]
*Azadirachta indica*	Chloroform extract of the aerial parts	Wistar rats, STZ	200 mg/kg	30 days	↓ Renal TBARS, AGEs ↓ Methylglyoxal and glycolaldehyde	[148]
*Abroma augusta*	Methanol extract of the leaves	Wistar rats, STZ-nicotinamide	100 mg/kg, 200 mg/kg	28 days	↓ Blood glucose ↓ Renal ROS production, and TBARS ↑ Renal GSH, CAT, SOD, GST, GPx, G6PD and GR ↓ Renal NF-κB and PKC isoforms ↑ Bcl-2 ↓ Renal Bax, caspase 3, and caspase 9 ↓ Renal IL-1β, IL-6, and TNF-α ↓ Renal damage	[149]
*Fragaria x ananassa*	Aqueous extract of the leaves	Wistar rats, STZ	50 mg/kg, 100 mg/kg, 200 mg/kg	30 days	↓ Blood glucose ↑ Plasma albumin and uric acid ↓ Urea nitrogen, creatinine, and Kim-1 ↓ Renal MDA ↑ Renal CAT and SOD ↓ Renal TNF-α and IL-6 ↓ Caspase-3 ↑ VEGF-A	[150]
*Quercus brantii*	Aqueous extract of the seeds	Wistar rats, STZ	100 mg/kg, 250 mg/kg, 500 mg/kg	21 days	↓ Blood glucose ↓ Serum creatinine and urea ↓ Renal MDA and GR ↑ Renal GSH, GST, CAT, GPx, SOD	[125]
*Cyclocarya paliurus*	Aqueous extract of the leaves	Wistar rats, HFD-STZ	47 mg/kg, 94 mg/kg	56 days	↓ Blood glucose ↓ Serum creatinine and BUN ↓ Serum IL-6 ↓ Renal index ↑ Renal CAT, GPx, SOD ↓ MDA ↓ Renal damage	[126]
*Ficus recemosa*	Ethanolic extract of the stem barks	Wistar rats, STZ	200 mg/kg, 400 mg/kg	56 days	↓ Blood glucose ↓ Serum creatinine and BUN ↑ Serum albumin ↓ Renal MDA ↑ Renal GSH, SOD ↓ Renal damage	[127]
*Punica granatum*	Methanol extract of the leaves	Wistar rats, STZ	100 mg/kg, 200 mg/kg, 400 mg/kg	56 days	↓ Blood glucose ↓ Serum creatinine and BUN ↑ Serum albumin ↓ Serum AGE ↓ Renal MDA ↑ Renal GSH, SOD, and CAT ↓ Renal damage	[151]
*Diplotaxis simplex*	Ethanolic extract of the flowers	Wistar rats, alloxan	100 mg/kg, 200 mg/kg	30 days	↓ Blood glucose ↓ Renal TBARS ↑ Renal SOD, CAT and GPx ↓ Serum urea and creatinine ↓ Renal damage	[128]
*Syzygium mundagam*	Methanolic extract of the barks	Wistar rats, STZ-nicotinamide	100 mg/kg, 200 mg/kg	28 days	↓ Blood glucose ↓ Serum urea ↓ Renal damage ↓ GR ↑ Renal SOD, CAT, GSH, GST	[129]
*Paederia foetida*	Methanolic extract of the leaves	Wistar rats, alloxan	250 mg/kg, 500 mg/kg	28 days	↓ Blood glucose ↓ Serum creatinine, BUN, TGF-β1 ↑ Creatinine clearance, albumin ↓ Renal MDA ↑ Renal GSH, SOD, and CAT ↓TNF-α, IL-6, IL-1β, NF-kB p65 ↓ Renal damage	[152]
*Lycium chinense*	Ethanolic extract of the leaves	Sprague–Dawley rats, STZ	100 mg/kg, 200 mg/kg, 400 mg/kg	40 days	↓ Renal MDA ↓ Serum creatinine, albumin, BUN, TGF-β1 ↑ Renal GSH, SOD, and CAT ↑ Creatinine clearance ↓ IL-1β, TNF-α, and IL-6 ↓ Renal damage	[153]
*Anogeissus acuminata*	Methanol extract of the leaves	Wistar rats, STZ	100 mg/kg, 300 mg/kg	56 days	↓ Blood glucose ↓ Serum MDA ↑ Serum CAT and GSH ↓ Urinary protein, serum creatinine, BUN	[130]
*Cassia auriculata*	Ethanolic extract of the buds and flowers	Wistar rats, HFD-STZ	250 mg/kg, 500 mg/kg	21 days	↓ Blood glucose ↓ Serum creatinine and urea ↓ Renal MDA ↑ Renal SOD, GPx, and GSH ↓ Renal damage	[131]
*Zingiber* *officinale*	Hydroethanolic (80%) extract of the rhizomes	Wistar rats, STZ	400 mg/kg, 800 mg/kg	42 days	↓ Blood glucose ↓ Renal MDA ↑ Renal SOD, CAT, and GSH ↓ Serum creatinine, urea, and BUN ↓ Urine albumin ↓ Renal damage ↓ TNF-α, IL-1β, and IL-6 ↓ Cytochrome c and caspase-3	[154]
*Allium cepa*	Hydroethanolic (80%) extract of the bulbs	Wistar rats, STZ	150 mg/kg, 300 mg/kg	28 days	↓ Blood glucose ↓ Serum uric acid, urea, AGEs ↓ Renal damage ↓ TNF-α, IL-1α, IL-10, collagen 1	[155]
*Acaciella angustissima*	Methanol extract of the pods	Wistar rats, STZ	25 mg/kg, 50 mg/kg, 100 mg/kg	28 days	↓ Blood glucose ↓ Urea, creatinine clearance, protein ↓ Renal TBARS	[124]
*Cassia obtusifolia*	Hydroethanolic extract of the seeds	Wistar rats, STZ	27 mg/kg, 54 mg/kg, 81 mg/kg	60 days	↓ Blood glucose ↓ IL-1β, IL-6 and TNF-α ↓ Serum MDA ↑ Serum SOD, CAT and GPx ↓ Urine protein, serum creatinine, and BUN ↓ Renal RAGE expression ↓ Renal damage	[156]
*Coriandrum sativum*	Petroleum ether extract of the seeds	Wistar rats, STZ-nicotinamide	100 mg/kg, 200 mg/kg, 400 mg/kg	45 days	↓ Blood glucose ↓ Urea, creatinine, BUN, and uric acid ↑ Creatinine clearance ↑ Renal SOD, GSH ↓ Renal MDA ↓ Renal AGEs ↓ Renal damage	[16]
*Nerium oleander*	Ethanolic extract of the stems and roots	Swiss albino mice, alloxan	200 mg/kg	20 days	↓ Blood glucose ↓ Serum creatinine, BUN, and uric acid ↑ Serum albumin ↑ Renal CAT, peroxidase ↓ Renal MDA	[132]
*Nelumbo nucifera*	Aqueous extract of the leaves	Sprague–Dawley rats, HFD-STZ	0.5% and 1% (*w*/*w*)	42 days	↓ Blood glucose ↓ Serum creatinine and BUN ↓ Renal damage ↓ Renal 8-OHdG and TBARS ↑ Renal SOD, CAT, GPx, and GSH	[119]
*Artemisia absinthium*	Ethanolic (70%) extract of the aerial parts	Wistar rats, STZ	200 mg/kg, 400 mg/kg	60 days	↓ Blood glucose ↓ Serum urea and creatinine ↑ Serum albumin ↑ Renal SOD ↓ Renal MDA ↓Renal TLR4, S100A4, and Bax genes expression ↑ Renal Bcl-2 gene expressions	[157]
*Scrophularia striata*	Ethanolic (70%) extract of the aerial parts	Wistar rats, STZ	100 mg/kg, 200 mg/kg	60 days	↓ Blood glucose ↓ Serum urea and creatinine ↑ Serum albumin ↑ Renal SOD ↓ Renal MDA ↓ Renal RAGE and S100A8 gene expressions ↓ Renal damage	[158]
*Gongronema latifolium*	Aqueous extract of the leaves	Wistar rats, alloxan	6.36 mg/kg, 12.72 mg/kg, and 25.44 mg/kg	13 days	↓ Blood glucose ↓ Serum uric acid, urea, creatinine, and BUN ↑ Renal SOD, CAT, GPx, GSH, GST ↓ Renal MDA ↓ IL-2 and IL-6	[159]
*Ficus religiosa*	Hydroethanolic extract of the aerial parts	Wistar rats, STZ-nicotinamide	50 mg/kg, 100 mg/kg, 200 mg/kg	45 days	↓ Blood glucose ↓ Serum creatinine, BUN, urea, and uric acid ↓ Renal TBARS ↑ Renal SOD, CAT, GSH ↓ Renal damage	[133]
*Phyllanthus fraternus*	Methanol extract of the aerial parts	Wistar rats, alloxan	200 mg/kg, 400 mg/kg	14 days	↓ Blood glucose ↓ Serum urea and creatinine ↓ Renal TBARS ↑ Renal GSH ↓ Renal damage	[134]
*Croton hookeri*	Methanol extract of the leaves	Sprague–Dawley, STZ	200 mg/kg	14 days	↓ Blood glucose ↓ Serum creatinine and BUN ↑ Renal damage ↓ Microalbumin ↓ KIM-1, NGAL, PKM2, SBP-1 ↓ Renal AGEs ↓ ROS, Renal MDA, and 8-OHdG ↑ Renal GSH, SOD, and CAT ↓ Renal IL-1β, IL-6, IL-10, and TGF-β1	[160]
*Thymelaea hirsuta*	Aqueous extract of the aerial parts	Wistar rats, STZ	200 mg/kg	28 days	↓ Blood glucose ↓ Urinary volume, glycosuria, and creatinine ↓ Tubulointerstitial renal collagen	[161]
*Anchomanes difformis*	Aqueous extract of the leaves	Wistar rats, fructose-STZ	200 mg/kg, 400 mg/kg	42 days	↓ Blood glucose ↓ Serum urea ↑ Renal CAT, SOD ↓ Renal IL-1β, IL-6, IL-10, IL-18 and TNF-α ↓ Renal NF-kB/p65 expression ↑ Renal Nrf2 expression ↑ Renal Bcl2 and caspase 3 ↓ Renal damage	[15]
*Allium jesdianum*	Ethanolic extract of the rhizomes	Wistar rats, STZ	250 mg/kg, 500 mg/kg	42 days	↓ Blood glucose ↓ Serum urea and creatinine ↑ Albumin ↓ Renal MDA ↑ Renal SOD ↓ CTGF and RAGE	[162]
*Lagerstroemia speciosa*	Extract of the leaves	Wistar rats, HFD-STZ	400 mg/kg	40 days	↓ Blood glucose ↓ Kidney hypertrophy ↓ BUN ↓ Serum creatinine ↓ MDA ↑ GSH ↓ TNF-α, IL-6, IL-1β ↓ AGEs and RAGE mRNA	[163]
*Trifolium alexandrinum*	Methanol extract of the aerial part	Wistar rats, HFD-STZ	200 mg/kg	35 days	↓ Serum urea, creatinine, BUN ↓ Urine volume, proteinuria ↓ TBARS ↑ Renal GSH, CAT ↓ TGF-β, TNF-α, and IL-6 ↓ GSK-3β ↓ Renal damage	[164]
*Etlingera elatior*	Aqueous extract of the flower	Sprague–Dawley rats HFD-STZ	1000 mg/kg	42 days	↓ Blood glucose, microalbuminuria, creatinine, BUN ↓ Plasma MDA ↑ Plasma SOD, CAT, GSH, T-AOC ↓ IL-6, TGF-β, CTGF ↓ Renal damage	[135]
*Centaurium erythraea*	Methanol extract of the aerial part	Wistar rats, STZ	100 mg/kg	28 days	↓ Serum creatinine, BUN ↓ Renal O-GlcNAc ↓ Renal TBARS ↑ Renal CAT, GPx, GR, MnSOD, CuZnSOD activities	[136]
*Thuja occidentalis*	Hydroalcoholic extracts of the aerial part	Wistar rats, STZ-nicotinamide	50 mg/kg, 100 mg/kg, 200 mg/kg	30 days	↓ Blood glucose, ↓ Serum urea, creatinine, BUN, uric acid ↓ Kidney index, renal AGEs, renal TBARs ↑ Renal SOD, CAT, GSH ↓ IL-6, TNF-α, TGF-β1	[137]

↑ indicates an increase, ↓ indicates a decrease.

### 6.2. Suppression of Inflammatory Mediators

Hyperglycemic-induced oxidative stress increases proinflammatory protein levels by invading macrophages. The inflammatory cytokines are then released, leading to local and systemic inflammation [165]. Excessive ROS production in pancreatic β-cells can activate stress signaling pathways, which in turn activate inflammatory and apoptotic transcription factors, including NF-κB. The result is the death of β-cells and a reduction in insulin. Most kidney cells, such as renal tubular, mesangial, glomerular endothelial, and dendritic cells, exhibit increased NF-κB expression in response to oxidative stress production [166,167,168]. Once NF-kB is activated, it triggers the transcription of proinflammatory genes encoding cytokines (TNFα, IL-1β, IL-2, IL-6, IL-12, and IL-18) and chemokines (MCP-1). The transcription of profibrotic genes involved in the production of growth factors (TGF-β) and leukocyte adhesion molecules (E-selectin, VCAM1, and ICAM-1) is also promoted by NF-κB. The production of these proinflammatory and profibrotic proteins results in inflammation, atherosclerosis, and vascular dysfunction. Thus, the approach targeting the inflammatory response may be effective for DN therapy [169]. An increased peroxisome proliferator-activated receptor-γ (PPARγ) expression has been found in diabetic animals. PPARγ is a ligand-activated transcription factor that belongs to the nuclear hormone receptor superfamily and is crucial for the control of cell cycle, insulin sensitivity, glucose, and lipid homeostasis. It is also involved in the development of diabetic kidney injury. Gao et al. [144] reported increased renal PPARγ expression and reduced renal damage after the administration of the ethanolic (70%) extract of the fruits of *Cornus officinalis* SIEB. et Zucc (100 mg/kg, 200 mg/kg or 400 mg/kg orally per day for 40 days) to diabetic Wistar rats induced by 60 mg/kg of STZ.

Medicinal plant extracts have been used to reduce the levels of proinflammatory mediators in animal models (Table 1). Honoré et al. [143] found that the administration of the water extract of the leaves of *Smallanthus sonchifolius* (70 mg/kg orally per day for 28 days) on diabetic Wistar rats induced by STZ (45 mg/kg) decreased the renal TGF-β1 expression. Lee et al. [145] showed that the administration of water extract of the of the aerial parts of *Portulaca oleracea* (300 mg/kg orally per day for 70 days) on C57BL/KsJ-db/db mice decreased the renal TGF-β1 level. The expression of NF-κB p65 and ICAM-1 in renal tissues was also observed. Lu et al. [146] revealed that the administration of aqueous–ethanol extract of the tuberous roots of *Liriope spicata* var. *prolifera* (100 mg/kg and 200 mg/kg orally per day for 56 days) to diabetic Wistar rats induced by STZ (60 mg/kg) decreased the renal expression of ICAM-1, MCP-1, TNF-α, IL-1β, and NF-κB. Studies on the effect of plant extracts on reducing the levels and expression of inflammatory mediators used the following treatments: methanolic extract of the leaves of *Paederia foetida* (250 mg/kg and 500 mg/kg orally per day for 28 days) administered to diabetic Wistar rats induced by 150 mg/kg of alloxan [152]; ethanolic extract of the leaves of *Lycium chinense* (100 mg/kg, 200 mg/kg, and 400 mg/kg orally per day for 40 days) administered to diabetic Sprague–Dawley rats induced by 65 mg/kg of STZ [153]; hydroethanolic extract of the seeds of *Cassia obtusifolia* (27 mg/kg, 54 mg/kg, and 81 mg/kg orally per day for 60 days) administered to diabetic Wistar rats induced by 40 mg/kg of STZ [156]; aqueous extract of the leaves of *Gongronema latifolium* (6.36 mg/kg, 12.72 mg/kg, and 25.44 mg/kg orally per day for 13 days) administered to diabetic Wistar rats induced by 150 mg/kg of alloxan [159]; methanolic extract of the leaves of *Croton hookeri* (200 mg/kg orally per day for 14 days) administered to diabetic Sprague–Dawley rats induced by 45 mg/kg of STZ [160]; and methanolic extract of the aerial part of *Trifolium alexandrinum* (200 mg/kg orally per day for 35 days) administered to diabetic Wistar rats induced by HFD and 35 mg/kg of STZ [164].

### 6.3. Inhibition of AGE Production

AGEs are lipids or proteins that have been glycated as a consequence of their interaction with glucose or related metabolites [170]. AGEs are generated through the Maillard process via a non-enzymatic reaction between ketones or aldehydes of reducing sugars and the terminal α-amino groups or ε-amino groups of protein lysine residues. The accumulation of AGEs at the site of microvascular injury plays a significant role in renal complication [171,172]. Cytotoxic AGEs may modify lipids and proteins, leading to renal inflammation and oxidative stress, both of which are hallmarks of diabetic kidney disease [173]. Diabetic rats having high AGE levels may be three times more likely to develop nephropathy compared with those having normal AGE levels [10]. AGE receptors are found in a variety of renal cells, including proximal tubular cells, mesangial cells, and podocytes. RAGE, LOX-1, galactin-3, CD-36, and SR-B1 are types of AGE receptors. AGEs increase inflammatory responses by activating and expressing various inflammatory mediators, such as IL-6, TGFβ1, and NF-κB [174].

Medicinal plant extracts show a potential therapeutic effect on the DN animal model via AGE suppression (Table 1). AGE production is inhibited in the kidney. Gutierrez and Ortiz [148] confirmed that the administration of *Azadirachta indica* chloroform extract (200 mg/kg orally per day for 30 days) to diabetic Wistar rats induced by STZ (50 mg/kg) inhibited the formation of AGEs. Other studies on the inhibition of AGE formation in diabetic rat models administered the following treatments: methanol extract of *Punica granatum* leaves (100 mg/kg, 200 mg/kg, and 400 mg/kg orally per day for 56 days) to diabetic Sprague–Dawley rats induced by 45 mg/kg of STZ [151]; hydroethanolic (80%) extract of *Allium cepa* bulbs (150 mg/kg and 300 mg/kg orally per day for 28 days) to diabetic Wistar rats induced by 50 mg/kg of STZ [155]; petroleum ether extract of *Coriandrum sativum* seeds (100 mg/kg, 200 mg/kg, and 400 mg/kg orally per day for 45 days) to diabetic Wistar rats induced by 65 mg/kg of STZ and 230 mg/kg of nicotinamide [16]; and *Lagerstroemia speciosa* leaf extract (400 mg/kg orally per day for 40 days) to diabetic Wistar rats induced by HFD and 35 mg/kg of STZ [163].

The interactions between AGEs and RAGEs trigger oxidative stress and promote the creation and release of cytokines, both of which amplify tissue damage [175]. Given the importance of the AGE–RAGEs axis in DN pathogenesis, inhibiting the formation of AGEs may be a promising treatment for DN. Lee et al. [141] found that administering the water extract of the roots of *Salvia miltiorrhiza* (501 mg/kg orally per day for 56 days) to Sprague–Dawley rats induced by STZ (45 mg/kg) decreased the level of AGEs (Table 1). The expression of glomerular RAGE (the receptor for AGEs) also decreased. The treatment of Wistar rats induced by STZ (60 mg/kg) with the hydroalcoholic extract of the roots of *Angelica Acutiloba* (50 mg/kg, 100 mg/kg, and 200 mg/kg orally per day for 56 days) reduced the overexpression of renal AGEs and RAGE [142]. Furthermore, the expression of renal RAGE increased in Wistar rats induced by STZ (55 mg/kg). However, this expression decreased after the administration of the ethanolic (70%) extract of the aerial parts *Scrophularia striata* at doses of 100 mg/kg and 200 mg/kg orally per day for 60 days [158]. High AGE levels in kidney tissues during diabetes can induce the secretion of CTGF, which has important roles in many biological processes. Diabetic rats with DN have high CTGF expression. Extracts that can suppress CTGF expression in the kidney may be useful to decrease the progression of DN. An increase in renal CTGF and RAGE expression levels was observed in diabetic Wistar rats induced by STZ (55 mg/kg) [162]. The treatment using the ethanolic extract of *Allium jesdianum* rhizomes at doses of 250 mg/kg and 500 mg/kg orally per day for 42 days decreased the renal CTGF and RAGE expression levels.

### 6.4. Suppression of Cells’ Apoptosis

Cell death is crucial for the development of DN. When renal tissues are subjected to oxidative stress over time, a variety of pathophysiological events may occur, ultimately resulting in cell death. Undesired cells are eliminated by apoptosis in normal tissues to maintain tissue homeostasis. When cells are damaged due to oxidative stress, apoptosis occurs and activates cell death receptors such as TNFRs [176,177]. Some pro- and anti-apoptotic proteins and cysteinyl aspartic acid-specific proteases (caspases) are the primary executors of apoptotic pathways [178]. Bcl-2 (anti-apoptotic) proteins act on mitochondria to regulate cytochrome c release and initiate the caspases-dependent apoptotic pathway [179]. In normal animal model, the expression of Bcl2, an anti-apoptotic protein, is mostly found in proximal and distal tubules and capsular parietal cells [180]. Bax is a pro-apoptotic protein that can modulate pro-apoptotic processes by inhibiting the expression of Bcl-2 members. In diabetic animals, caspase-3 and caspase-9 are upregulated in renal tissues. The activation of Bax inhibits Bcl-2, indicating the occurrence of apoptotic cell damage in renal tissues with the progression of DN [181].

As seen in Table 1, Khanra et al. [149] administered the methanol extract of the leaves of *Abroma augusta* at doses of 100 and 200 mg/kg orally per day for 28 days to diabetic rats induced by STZ (65 mg/kg) and nicotinamide (110 mg/kg) and found that the expression of renal Bcl-2 increased and that of renal Bax, caspase 3, and caspase 9 decreased. In diabetic Wistar rats induced by 55 mg/kg of STZ, the administration of the ethanolic extract of the aerial parts of *Artemisia absinthium* (200 mg/kg and 400 mg/kg orally per day for 60 days) decreased the expression of renal Bax and eventually increased the expression of renal Bcl-2 [157]. Alabi et al. [15] observed an increase in renal Bcl2 and caspase-3 expression levels on diabetic rats induced by fructose 10% and STZ (40 mg/kg). The administration of the water extract of *Anchomanes difformis* at doses of 200 mg/kg and 400 mg/kg orally per day for 42 days also produced this result.

Podocytes secret vascular endothelial growth factor (VEGF), which is required for the survival of endothelial cells, podocytes, and mesangial cells. The expression of anti-apoptotic protein Bcl-2 is also increased by VEGF [182]. Ibrahim and Abd El-Maksoud [150] showed that the administration of the water extract of *Fragaria x ananassa* leaves (50 mg/kg, 100 mg/kg, and 200 mg/kg orally per day for 30 days) to diabetic Wistar rats induced by STZ (45 mg/kg) increased the level of renal VEGF-A and decreased the activity of caspase 3 (Table 1). The production of ROS in mitochondria can result in the release of cytochrome c into the cytosol, leading to caspase 3 activation and apoptotic cell death. The administration of the hydroethanolic extract of the rhizomes of *Zingiber officinale* (400 mg/kg and 800 mg/kg orally per day for 42 days) to diabetic Wistar rats induced by STZ (50 mg/kg) reduced the levels of renal cytochrome c and caspase-3 [154].

### 6.5. Regulation of Tissue Injury- and Renal Fibrosis-Related Protein

Nephrin is a crucial transmembrane protein located in the slit diaphragm complex. It serves as the framework for the podocyte slit diaphragm and is involved in podocyte survival [183,184]. Through podocin and CD2-associated protein (CD2AP), nephrin is connected to the actin cytoskeleton [185,186]. This compound is crucial for controlling the insulin sensitivity of podocytes because its cytoplasmic domain permits the docking of GLUT1 and GLUT4 with the vesicle-associated membrane protein-2, which facilitates insulin signaling. It can act as an early diabetic kidney-disease biomarker. Damage to podocytes has been connected to modifications in nephrin excretion [187,188]. Podoclin, a protein, helps form tight connections between the foot processes of podocytes. Its levels in urine can be measured to monitor the development of kidney damage in people with diabetes [189]. DN affects nephrin and podocin expression. Therefore, DN progression might be slowed down by upregulating these proteins. Tzeng et al. [147] reported that the administration of the ethanolic extract of *Zingiber zerumbet* (200 mg/kg and 300 mg/kg orally per day for 56 days) to diabetic Wistar rats induced by STZ (60 mg/kg) increased the expression levels of renal nephrin and podocin (Table 1).

Extracellular matrix (ECM) protein builds up in the tubulointerstitial space and the glomerular mesangium of DN, which results in renal fibrosis and ultimately renal failure [190,191]. The ECM is a highly charged, dynamic structure that participates actively in cell signaling and serves as a support system for the cells. It is made up of molecules of elastin, glycoproteins, and collagen that interact with one another and the cells around them to form a complex network [192]. Renal fibrosis is a pathogenic intermediary for chronic kidney disease (CKD) progression and a histological marker of CKD [193]. Abid et al. [161] showed that the administration of the water extract of the aerial parts of *Thymelaea hirsute* (at a dose of 200 mg/kg orally per day for 28 days) to diabetic Wistar rats induced by STZ (50 mg/kg) decreased the levels of urinary creatinine and tubulointerstitial renal collagen (Table 1). This result demonstrates that the extract could protect against renal fibrosis by inhibiting ECM protein accumulation.

## 7. Conclusions

DN, often known as diabetic kidney disease, refers to kidney dysfunction caused by DM. The exceedingly complicated and poorly understood causes of DN have rendered its treatment challenging and expensive. A different approach to the treatment of DN may involve extracts from medicinal plants, which have the ability to slow the advancement of this condition. This activity is closely related to the bioactive content of medicinal plant extracts, especially the phenolic and flavonoid groups. By decreasing oxidative stress; suppressing inflammation, AGE generation, cell apoptosis; and controlling tissue injury-related protein expression, medicinal plant extracts can slow the development of DN. This review is limited to the use of crude extracts of various medicinal plants. However, the presence of various phytochemical compounds in extract may provide its own advantages because of the interaction between these compounds. In addition, clinical trials need to be performed to have a better knowledge about the efficacy of these medicinal plant extracts to human.

## Figures and Tables

**Figure 1 life-13-00560-f001:**
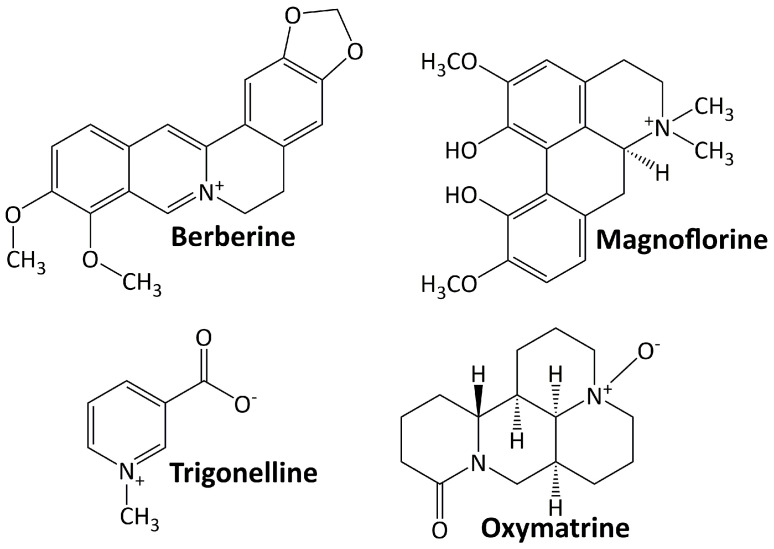
Alkaloid compounds targeting diabetic nephropathy.

**Figure 2 life-13-00560-f002:**
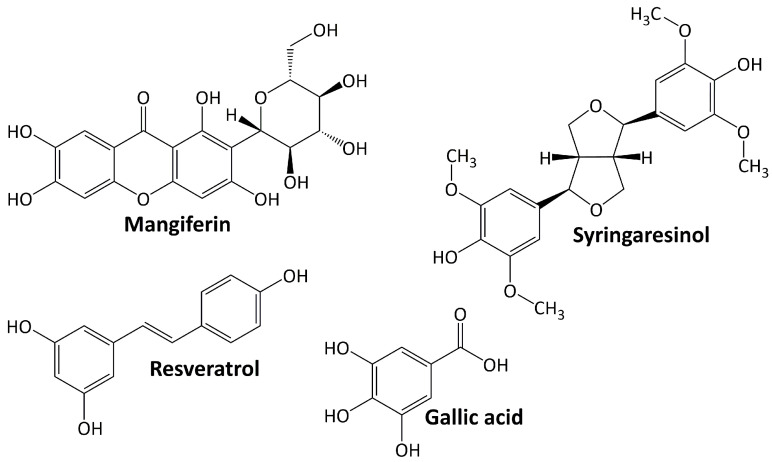
Phenolic compounds targeting diabetic nephropathy.

**Figure 3 life-13-00560-f003:**
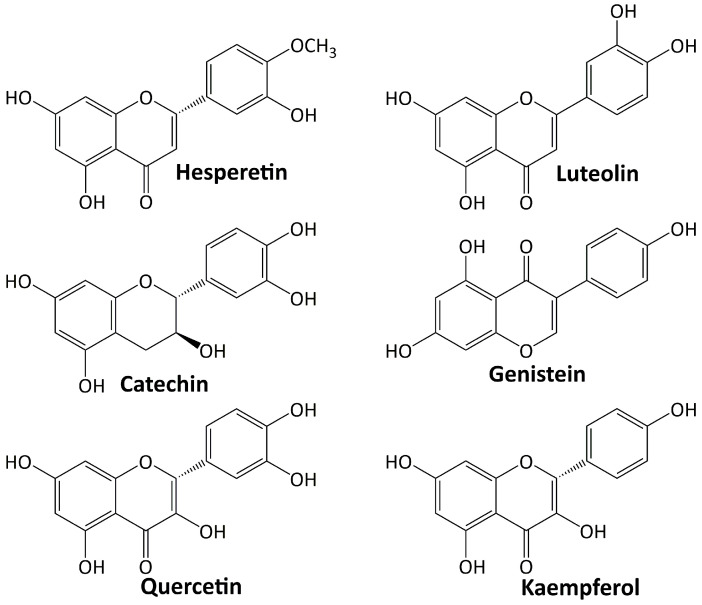
Flavonoid compounds targeting diabetic nephropathy.

**Figure 4 life-13-00560-f004:**
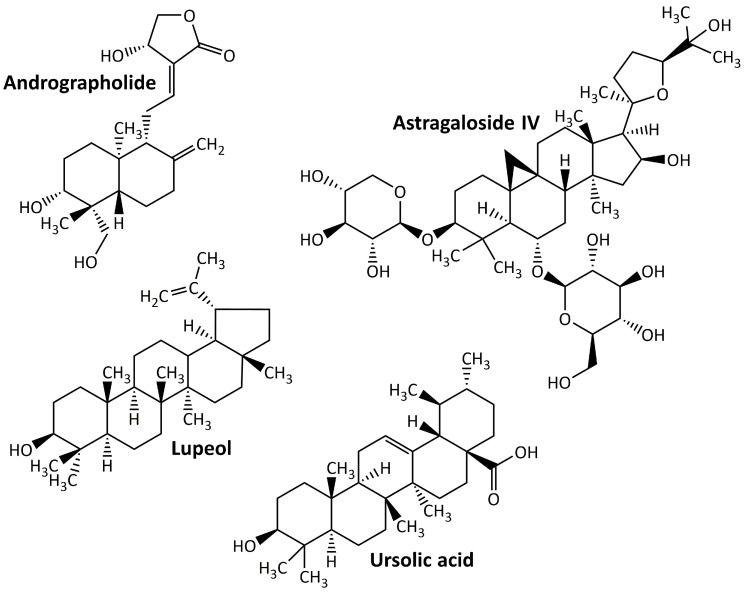
Terpenoid compounds targeting diabetic nephropathy.

**Figure 5 life-13-00560-f005:**
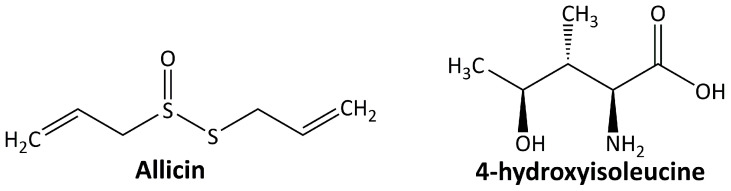
Miscellaneous compounds targeting diabetic nephropathy.

**Figure 6 life-13-00560-f006:**
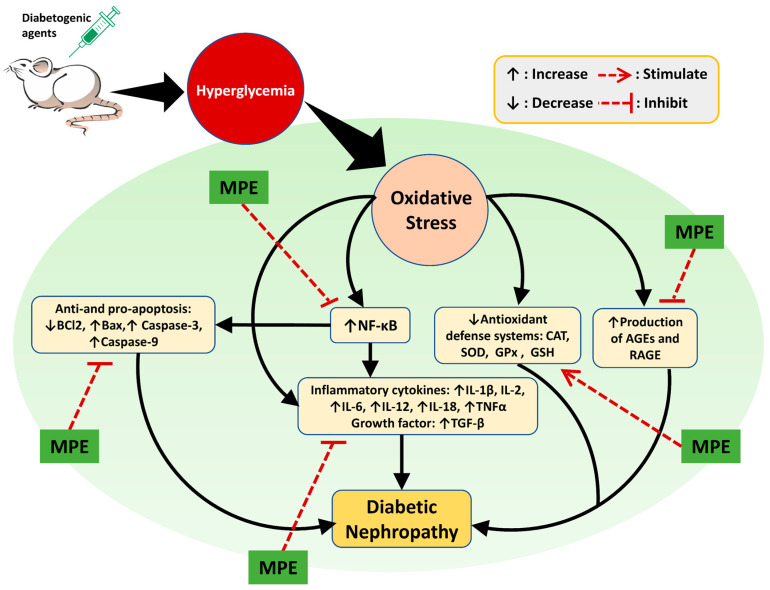
Possible mechanisms of action of plant extracts in DN. Experimental animal were induced using diabetogenic agents such as STZ, alloxan, and HFD causing hyperglycemia which in turn causes oxidative stress. Oxidative stress in the kidney triggers the occurrence of DN through increased inflammation, apoptosis, and production of AGEs. In addition, decreased levels of antioxidants can also cause DN. Oxidative stress can increase the expression of NF-kB in the kidney so that it activates proinflammatory cytokines (IL-1β, IL-2, IL-6, IL-12, IL-18, and TNFα) and produces growth factors (TGF-β). NF-κB also reduces the anti-apoptosis (Bcl2) factor and increases the pro-apoptosis (Bax, Caspase-3, and Caspase-9) factors. Decreased renal antioxidant levels (SOD, CAT, GPx, and GSH) as a result of oxidative stress exacerbate DN progression since they can no longer protect the kidney from the adverse effects of ROS. An increase in the number of AGEs in kidney leads to inflammation as a result of the modification of lipids and proteins. Medicinal plant extracts (MPE) show a renoprotective effect on the DN animal model via amelioration of the abovementioned causative factors.

## Data Availability

Not applicable.

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
