# Peer review of "A Review of Medicinal Plants with Renoprotective Activity in Diabetic Nephropathy Animal Models"

_life, 2023, doi:10.3390/life13020560_

Round 1

Reviewer 1 Report

life-2160115

Delete All Simple Summary lines 16-27

line 203 forming growth factor change as  forming growth factor

line 269 signal transducer and activator of transcription  change as signal transducer and activator of transcription

lines 307 to 308 and has shown  change as and has shown  change

Reviewer 2 Report

Major comments:

Methodology:

·         The methodology of the study including search strategy, articles selection, data extraction, data analysis and statistical analysis was not mentioned. Please add in details.

·         Please add results section.

·         In the discussion, the pathomechanism was mentioned collectively at the end and was not linked to each medicinal plant which renders it vague and inaccurate. It would be nice to add the pathomechanism of each medicinal plant separately.

Minor comments:

·         The article needs thorough language revision.

·         Abstract :

·         It would be better to remove “also known as high blood sugar level” line 30 as it is basic knowledge.

·         Line 31: “DN can promote chronic renal disease and kidney failure.” DN is itself a type of nephropathy, the sentence is not accurate.

·         Line 48 “As a result of DN, the kidney become increasingly ineffective at removing toxins and drugs from the body”. The sentence is too basic, please remove.

Reviewer 3 Report

This review discussed the importance of plant extracts in the treatment of diabetic neuropathy using animal models. Overall, it's a comprehensive review, but there is a lack of references for certain statements throughout the manuscript. Some information is not summarised in a way its more understandable. Revisions will improve the overall content and status of the review.

Introduction 63-70 lines

This paragraph should be revised to include the review's aims instead of discussing the methods. Instead, it may be good to have a separate section under a subheading to mention the approach of the review.

Line 75: What are you referring to as "This condition"

Under "3. Animal Model of DN", there are a lot of descriptions of various diabetic agents, and it will be clearer if they are under un headings instead of paragraphs under the same heading "3. Animal Model of DN".

Line 157: in what animal model and mention the reference?

Line: 177: They are present in 40% of 176 plant families - insert the reference

Line 181-182: A total of 12,000 plant alkaloids with various medical applications have been studied [45]. Are there any clinical trials in those studies? Are you talking about both pre-clinal and clinical trials?

The figures have not been discussed in the text.

Line 187-209: break down the paragraph to discuss the different alkaline instead of discussing them under the same sections. You may use sub subheadings.

Line 210-338: The same comment also applies to other compounds.

Line 339- "Renoprotective Mechanism of Medicinal Plants" should be changed to Renoprotective Mechanisms of Medicinal Plants as you are discussing several mechanisms.

Line 348: Figure 6 should be described briefly as a legend.

Line 374-428: This section can be given as a table for easy reading. When discussing the same information in the table, refer to the table.

The same comment applies to other mechanisms as well. It feels very overwhelming with lots of information. Use tables and diagrams where necessary for easing reading.

Discussion is missing. What are the limitations and strengths of the review? Implications and future direction should be discussed.

Round 2

Reviewer 2 Report

The manuscript is appropriate in its current form.

Author Response

Reviewer 2 comment: The manuscript is appropriate in its current form.

Author's response: We would like to express our gratitude for your insightful comment which have tremendously assisted us in improving the quality of our article.

Reviewer 3 Report

This review discussed the importance of plant extracts in treating diabetic neuropathy using animal models. Overall, it’s a comprehensive review, and the authors have addressed the comments satisfactorily.  

Author Response

Reviewer 3 comment: This review discussed the importance of plant extracts in treating diabetic neuropathy using animal models. Overall, it’s a comprehensive review, and the authors have addressed the comments satisfactorily

Author's comment: We would like to thank the Reviewers, which have tremendously assisted us in improving the quality of our article.